# TIME-AGNOSTIC PREDICTION: PREDICTING PREDICTABLE VIDEO FRAMES

**Dinesh Jayaraman**
UC Berkeley

**Frederik Ebert**
UC Berkeley

**Alyosha Efros**
UC Berkeley

**Sergey Levine**
UC Berkeley

## ABSTRACT

Prediction is arguably one of the most basic functions of an intelligent system. In general, the problem of predicting events in the future or between two waypoints is exceedingly difficult. However, most phenomena naturally pass through relatively predictable bottlenecks—while we cannot predict the precise trajectory of a robot arm between being at rest and holding an object up, we *can* be certain that it must have picked the object up. To exploit this, we decouple visual prediction from a rigid notion of time. While conventional approaches predict frames at regularly spaced temporal intervals, our time-agnostic predictors (TAP) are not tied to specific times so that they may instead discover predictable "bottleneck" frames no matter when they occur. We evaluate our approach for future and intermediate frame prediction across three robotic manipulation tasks. Our predictions are not only of higher visual quality, but also correspond to coherent semantic subgoals in temporally extended tasks.

## 1 INTRODUCTION

Imagine taking a bottle of water and laying it on its side. Consider what happens to the surface of the water as you do this: which times can you confidently make predictions about? The surface is initially flat, then becomes turbulent, until it is flat again, as shown in Fig 1. Predicting the exact shape of the turbulent liquid is extremely hard, but its easy to say that it will eventually settle down.

Prediction is thought to be fundamental to intelligence (Bar, 2009; Clark, 2013; Hohwy, 2013). If an agent can learn to predict the future, it can take anticipatory actions, plan through its predictions, and use prediction as a proxy for representation learning. The key difficulty in prediction is uncertainty. Visual prediction approaches attempt to mitigate uncertainty by predicting iteratively in heuristically chosen small timesteps, such as, say, $0.1s$. In the bottle-tilting case, such approaches generate blurry images of the chaotic states at $t = 0.1s, 0.2s, \ldots$, and this blurriness compounds to make predictions unusable within a few steps. Sophisticated probabilistic approaches have been proposed to better handle this uncertainty (Babaeizadeh et al., 2018; Lee et al., 2018; Denton & Fergus, 2018; Xue et al., 2016).

What if we instead change the goal of our prediction models? Fixed time intervals in prediction are in many ways an artifact of the fact that cameras and monitors record and display video at fixed frequencies. Rather than requiring predictions at regularly spaced future frames, we ask: if a frame prediction is treated as a bet on that frame occurring at *some* future point, what should we predict? Such time-agnostic prediction (TAP) has two immediate effects: (i) the predictor may skip more uncertain states in favor of less uncertain ones, and (ii) while in the standard approach, a prediction is wrong if it occurs at $t \pm \epsilon$ rather than at $t$, our formulation considers such predictions equally correct.

Recall the bottle-tilting uncertainty profile. Fig 1 depicts uncertainty profiles for several other prediction settings, including both forward/future prediction (given a start frame) and intermediate prediction (given start and end frames). Our time-agnostic reframing of the prediction problem targets the minima of these profiles, where prediction is intuitively easiest. We refer to these minima states as "bottlenecks."

At this point, one might ask: are these "easy" bottlenecks actually useful to predict? Intuitively, bottlenecks naturally correspond to reliable subgoals—an agent hoping to solve the maze in Fig 1 (e) would do well to target its bottlenecks as subgoals. In our experiments, we evaluate the usefulness of our predictions as subgoals in simulated robotic manipulation tasks.

Figure 1: (a) Over time as the bottle is tilted, the uncertainty first rises and then falls as the bottle is held steady after tilting. (b)-(e) Similar uncertainty profiles corresponding to various scenarios—a ball rolling down the side of a bowl, a car driving on a highway with an exit 100m away, an iron pellet tossed in the direction of a magnet, and intermediate frame prediction in a maze traversal given start and end states. The red asterisks along the x-axis correspond to the asterisks in the maze—these "bottleneck" states *must* occur in any successful traversal.

Our main contributions are: (i) we reframe the video prediction problem to be time-agnostic, (ii) we propose a novel technical approach to solve this problem, (iii) we show that our approach effectively identifies "bottleneck states" across several tasks, and (iv) we show that these bottlenecks correspond to subgoals that aid in planning towards complex end goals.

## 2  RELATED WORK

**Visual prediction approaches.**   Prior visual prediction approaches regress directly to future video frames in the pixel space (Ranzato et al., 2014; Oh et al., 2015) or in a learned feature space (Hadsell et al., 2006; Mobahi et al., 2009; Jayaraman & Grauman, 2015; Wang et al., 2016; Vondrick et al., 2016b; Kitani et al., 2012). The success of generative adversarial networks (GANs) (Goodfellow et al., 2014; Mirza & Osindero, 2014; Radford et al., 2015; Isola et al., 2017) has inspired many video prediction approaches (Mathieu et al., 2015; Vondrick et al., 2016a; Finn & Levine, 2017; Xue et al., 2016; Oh et al., 2015; Ebert et al., 2017; Finn et al., 2016; Larsen et al., 2016; Lee et al., 2018). While adversarial losses aid in producing photorealistic image patches, prediction has to contend with a more fundamental problem: uncertainty. Several approaches (Walker et al., 2016; Xue et al., 2016; Denton & Fergus, 2018; Lee et al., 2018; Larsen et al., 2016; Babaeizadeh et al., 2018) exploit conditional variational autoencoders (VAE) (Kingma & Welling, 2013) to train latent variable models for video prediction. Pixel-autoregression (Oord et al., 2016; van den Oord et al., 2016; Kalchbrenner et al., 2016) explicitly factorizes the joint distribution over all pixels to model uncertainty, at a high computational cost.

Like these prior approaches, we too address the uncertainty problem in video prediction. We propose a general time-agnostic prediction (TAP) framework for prediction tasks. While all prior work predicts at fixed time intervals, we aim to identify inherently low-uncertainty bottleneck frames with no associated timestamp. We show how TAP may be combined with conditional GANs as well as VAEs, to handle the residual uncertainty in its predictions.

**Bottlenecks.**   In hierarchical reinforcement learning, bottlenecks are proposed for discovery of options (Sutton et al., 1999) in low-dimensional state spaces in (McGovern & Barto, 2001; Şimşek & Barto, 2009; Bacon, 2013; Metzen, 2013). Most approaches (Şimşek & Barto, 2009; Bacon, 2013; Metzen, 2013) construct full transition graphs and apply notions of graph centrality to locate bottlenecks. A multi-instance learning approach is applied in (McGovern & Barto, 2001) to mine states that occur in successful trajectories but not in others. We consider the use of our bottleneck predictions as subgoals for a hierarchical planner, which is loosely related to options in that both aim to break down temporally extended trajectories into more manageable chunks. Unlike these prior works, we use *predictability* to identify bottlenecks, and apply this to unlabeled high-dimensional visual state trajectories.

Concurrently with us, Neitz et al. (2018) also propose a similar idea that allows a predictor to select when to predict, and their experiments demonstrate its advantages in specially constructed tasks with clear bottlenecks. In comparison, we propose not just the basic time-agnostic loss (Sec 3.1), but also improvements in Sec 3.2 through 3.5 that allow time-agnostic prediction to work in more general tasks such as synthetic and real videos of robotic object manipulation. Our experiments also test the quality of discovered bottlenecks in these scenarios and their usefulness as subgoals for hierarchical planning.

## 3 TIME-AGNOSTIC PREDICTION OF BOTTLENECK FRAMES

In visual prediction, the goal is to predict a set of unobserved target video frames given some observed context frames. In forward prediction, the context is the first frame, and the target is all future frames. In the bidirectionally conditioned prediction case, the context is the first and the last frame, and the frames in between are the target. In Fig 1, we may wish to predict future images of the tilting bottle, or intermediate images of an agent who traverses the maze successfully.

### 3.1 MINIMUM-OVER-TIME LOSS

In standard fixed-time video prediction models (Ranzato et al., 2014; Oh et al., 2015; Mathieu et al., 2015; Vondrick et al., 2016a; Walker et al., 2016; Finn & Levine, 2017; Xue et al., 2016; Oh et al., 2015; Ebert et al., 2017; Finn et al., 2016; Lee et al., 2018; Denton & Fergus, 2018), a frame $x_\tau$ (video frame at time $\tau$) is selected in advance to be the training target for some given input frames $c$. For instance, in a

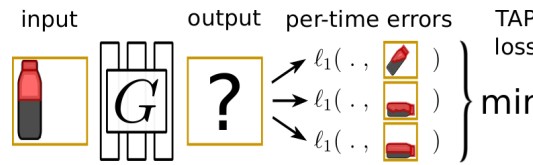

Figure 2: The TAP minimum-over time loss.

typical forward prediction setup, the input may be $c = x_0$, and the target frame may be set to $x_\tau = x_1$. A predictor $G$ takes context frames $c$ as input and produces a single frame $G(c)$. $G$ is trained as:

$$G^* = \arg\min_G \mathcal{L}_0(G) = \arg\min_G \mathcal{E}(G(c), x_\tau), \tag{1}$$

where $\mathcal{E}$ is a measure of prediction error, such as $\|G(c) - x_\tau\|_1$.[1] This predictor may be applied recursively at test time to generate more predictions as $G(G(c))$, $G(G(G(c)))$, and so on.

We propose to depart from this fixed-time paradigm by decoupling prediction from a rigid notion of time. Instead of predicting the video frame at a specified time $\tau$, we propose to predict predictable *bottleneck* video frames through a time-agnostic predictor (TAP), as motivated in Sec 1. To train this predictor, we minimize the following "minimum-over-time" loss:

$$G^* = \arg\min_G \mathcal{L}(G) = \arg\min_G \min_{t \in T} \mathcal{E}(G(c), x_t), \tag{2}$$

where the key difference from Eq 1 is that the loss is now a minimum over target times of a time-indexed error $\mathcal{E}_t = \mathcal{E}(\cdot, x_t)$. The target times are defined by a set of time indices T. For forward prediction starting from input $c = x_0$, we may set targets to $T = \{1, 2, ...\}$. Fig 2 depicts this idea schematically. Intuitively, the penalty for a prediction is determined based on the closest matching ground truth target frame. This loss incentivizes the model to latch on to "bottleneck" states in the video, *i.e.*, those with low uncertainty. In the bottle-tilting example, this would mean producing an image of the bottle after the water has come to rest.

One immediate concern with this minimum-over-time TAP loss might be that it could produce degenerate predictions very close to the input conditioning frames $c$. However, as in the tilting bottle and other cases in Fig 1, uncertainty is not always lowest closest to the observed frames. Moreover, target frame indices T are always disjoint from the input context frames, so the model's prediction must be different from input frames by at least one step, which is no worse than the one-step-forward prediction of Eq 1. In our experiments, we show cases where the minimum-over-time loss above captures natural bottlenecks successfully. Further, Sec 3.2 shows how it is also possible to explicitly penalize predictions near input frames $c$.

This minimum loss may be viewed as adaptively learning the time offset $\tau$, but in fact, the predictor's task is even simpler since it is not required to provide a timestamp accompanying its prediction. For example, in Fig 1(e), it need only specify which points in the maze the agent will go through; it need not specify *when*. Lifting the requirement of a timestamped prediction relieves TAP approaches of a significant implicit burden.

**Recursive TAP.** TAP models may also be trained for recursive prediction, by minimizing the following loss:

$$G^* = \arg\min_G \mathcal{L}_{\text{rec}}(G) = \arg\min_G \sum_r \min_{t \in T(r)} \mathcal{E}(G(c(r)), x_t), \tag{3}$$

---

[1]We omit expectations over training samples throughout this paper to avoid notational clutter.

where $c(r)$ and $\mathrm{T}(r)$ are the input and target set at recursion level $r$, both dynamically adapted based on the previous prediction. The input $c(r)$ may be set to the previous prediction $G(c(r-1))$, so that the sequence of predictions is $(G(c(0)), G(G(c(0))), \ldots)$. $\mathrm{T}(r)$ is set to target all times *after* the last prediction. In other words, if the prediction at $r = 0$ was closest to frame $x_5$, the targets for $r = 1$ are set to $\mathrm{T}(1) = \{6, 7, \ldots\}$. While we also test recursive TAP in Sec 4, in the rest of this section, we discuss the non-recursive formulation, building on Eq 2, for simplicity.

**Bidirectional TAP.** Finally, while the above description of TAP has focused on forward prediction, the TAP loss of Eq 2 easily generalizes to bidirectional prediction. Given input frames $c = (x_0, x_{\text{last}})$, fixed-time bidirectional predictors might target, say, the middle frame $x_\tau = x_{\text{last}/2}$. Instead, bidirectional TAP models target all intermediate frames, i.e., $\mathrm{T} = \{1, 2, \ldots, \text{last} - 1\}$ in Eq 2. As in forward prediction, the model has incentive to predict predictable frames. In the maze example from Fig 1, this would mean producing an image of the agent at one of the asterisks.

## 3.2 FROM MINIMUM TO GENERALIZED MINIMUM TAP LOSS

Within the time-agnostic prediction paradigm, we may still want to specify preferences for some times over others, or for some visual properties of the predictions. Consider the minimum-over-time loss $\mathcal{L}$ in Eq 2. Taking the minimum inside, this may be rewritten as:

$$\mathcal{L}(G) = \min_{t \in T} \mathcal{E}_t = \mathcal{E}_{\arg\min_{t \in T} \mathcal{E}_t}, \tag{4}$$

where we use the time-indexed error $\mathcal{E}_t$ as shorthand for $\mathcal{E}(., x_t)$. We may now extend this to the following "generalized minimum" loss, where the outer and inner errors are decoupled:

$$\mathcal{L}'(G) = \mathcal{E}_{\arg\min_{t \in T} \mathcal{E}'_t}. \tag{5}$$

Now, $\mathcal{E}'_t$, over which the minimum is computed, could be designed to express preferences about which frames to predict. In the simplest case, $\mathcal{E}'_t = \mathcal{E}_t$, and the loss reduces to Eq 2. Instead, suppose that predictions at some times are preferred over others. Let $w(t)$ express the preference value for all target times $t \in T$, so that higher $w(t)$ indicates higher preference. Then we may set $\mathcal{E}'_t = \mathcal{E}_t / w(t)$ so that times $t$ with higher $w(t)$ are preferred in the $\arg\min$. In our experiments, we set $w(t)$ to linearly increase with time during forward prediction and to a truncated discrete Gaussian centered at the midpoint in bidirectional prediction.

At this point, one might ask: could we not directly incorporate preferences into the outer error? For instance, why not simply optimize $\min_t \mathcal{E}_t / w(t)$? Unfortunately, that would have the side-effect of *downweighting* the errors computed against frames with higher preferences $w(t)$, which is counterproductive. Decoupling the outer and inner errors instead, as in Eq 5, allows applying preferences $w(t)$ only to select the target frame to compute the outer loss against; the outer loss itself penalizes prediction errors equally regardless of which frame was selected.

The generalized minimum formulation may be used to express other kinds of preferences too. For instance, when using predictions as subgoals in a planner, perhaps some states are more expensive to reach than others. We also use the generalized minimum to select frames using different criteria than the prediction penalty itself, as we will discuss in Sec 3.5.

## 3.3 TIME-AGNOSTIC CONDITIONAL GANS

TAP is not limited to simple losses such as $\ell_1$ or $\ell_2$ errors; it can be extended to handle expressive GAN losses to improve perceptual quality. A standard conditional GAN (CGAN) in fixed-time video prediction targeting time $\tau$ works as follows: given a "discriminator" $D$ that outputs 0 for input-prediction tuples and 1 for input-ground truth tuples, the generator $G$ is trained to fool the discriminator. The discriminator in turn is trained adversarially using a binary cross-entropy loss. The CGAN objective is written as:

$$G^* = \arg\min_{G} \ \max_{D} \mathcal{L}_{\text{cgan}}(G, D),$$
$$\mathcal{L}_{\text{cgan}}(G, D) = \log(D(c, x_\tau)) + \log(1 - D(c, G(c)) \tag{6}$$

To make this compatible with TAP, we train $|T|$ discriminators $\{D_t\}$, one per timestep. Then, analogous to Eq 2, we may define a time-agnostic CGAN loss:

$$G^* = \arg\min_G \min_{t \in T} \max_{D_t} \mathcal{L}_{\text{cgan}}^t(G, D_t),$$

$$\mathcal{L}_{\text{cgan}}^t(G, D_t) = \log D_t(c, x_t) + \log(1 - D_t(c, G(c))) + \sum_{t' \neq t} \log(1 - D_t(c, x_{t'})), \qquad (7)$$

Like Eq 6, Eq 7 defines a cross-entropy loss. The first two terms are analogous to Eq 6 — for the $t$-th discriminator, the $t$-th frame provides a positive, and the generated frame provides a negative instance. The third term treats ground truth video frames occurring at other times $x_{t' \neq t}$ as negatives. In practice, we train a single discriminator network with $|T|$ outputs that serve as $\{D_t\}$. Further, for computational efficiency, we approximate the summation over $t' \neq t$ by sampling a single frame at random for each training video at each iteration. Appendix A provides additional details.

### 3.4 TIME-AGNOSTIC CONDITIONAL VAEs

While TAP targets low-uncertainty bottleneck states, it may be integrated with a conditional variational autoencoder (CVAE) to handle residual uncertainty at these bottlenecks. In typical fixed-time CVAE predictors targeting time $\tau$, variations in a latent code $z$ input to a generator $G(c, z)$ must capture stochasticity in $x_\tau$. At training time, $z$ is sampled from a posterior distribution $q_\phi(z|x_\tau)$ with parameters $\phi$, represented by a neural network. At test time, $z$ is sampled from a prior $p(z)$. The training loss combines a log-likelihood term with a KL-divergence from the prior:

$$\mathcal{L}_{\text{cvae}}(G, \phi) = D_{KL}(q_\phi(z|x_\tau), p(z)) - \mathbb{E}_{z \sim q_\phi(z|x_\tau)} \ln p_G(x_\tau | c, z), \qquad (8)$$

where we might set $p_G$ to a Laplacian distribution such that the second term reduces to a $l1$-reconstruction loss $-\ln p_G(x_\tau|c, z) = \|G(c, z) - x_\tau\|_1$. In a time-agnostic CVAE, rather than capturing stochasticity at a fixed time $\tau$, $z$ must now capture stochasticity at bottlenecks: e.g., when the agent crosses one of the asterisks in the maze of Fig 1, which *pose* is it in? The bottleneck's time index varies and is not known in advance. For computational reasons (see Appendix B), we pass the entire video $X$ into the inference network $q_\phi$, similar to Babaeizadeh et al. (2018). The negative log-likelihood term is adapted to be a minimum-over-time:

$$\mathcal{L}_{\text{cvae}}(G, \phi) = D_{KL}(q_\phi(z|X), p(z)) + \min_{t \in T} \mathbb{E}_{z \sim q_\phi(z|X)} \left[ -\ln p_G(x_t|c, z) \right]. \qquad (9)$$

### 3.5 COMBINED LOSS, NETWORK ARCHITECTURE, AND TRAINING

We train time-agnostic CVAE-GANs with the following combination of a generalized minimum loss (Sec 3.2) and the CVAE KL divergence loss (Sec 3.4):

$$G^* = \arg\min_G \min_\phi \left[ D_{KL}(q_\phi(z|X), p(z)) + \mathcal{E}_{\arg\min_{t \in T} \mathcal{E}_t'} \right],$$

$$\mathcal{E}_t = \max_{D, D'} \mathcal{L}_{\text{cgan}}^t(G, D_t) + \mathcal{L}_{\text{cvae-gan}}^t(G, D_t') + \|G(c, z) - x_t\|_1,$$

$$\mathcal{E}_t' = \|G(c, z) - x_t\|_1 / w(t). \qquad (10)$$

The outer error $\mathcal{E}_t$ absorbs the CGAN discriminator errors (Sec 3.3), while the inner error $\mathcal{E}_t'$ is a simple $\ell_1$ error, weighted by user-specified time preferences $w(t)$ (Sec 3.2). Omitting GAN error terms in $\mathcal{E}_t'$ helps stabilize training, since learned errors may not always be meaningful especially early on in training. As in VAE-GANs (Larsen et al., 2016; Lee et al., 2018), the training objective includes a new term $\mathcal{L}_{\text{cvae-gan}}^t$, analogous to $\mathcal{L}_{\text{cgan}}^t$ (Eq 7). We set up $\mathcal{L}_{\text{cgan}}^t$ to use samples $z$ from the prior $p(z)$, while $\mathcal{L}_{\text{cvae-gan}}^t$ instead samples $z$ from the posterior $q_\phi(z|X)$, and employs separate discriminators $\{D_t'\}$. The $\ell_1$ loss also samples $z$ from the posterior. We omit expectations over the VAE latent $z$ to keep notation simple.

Frame generation in the predictor involves first generating appearance flow-transformed input frames (Zhou et al., 2016) and a frame with new uncopied pixels. These frames are masked and averaged to produce the output. Full architecture and training details are in Appendix C.

Grasping episode (length $T = 15$)

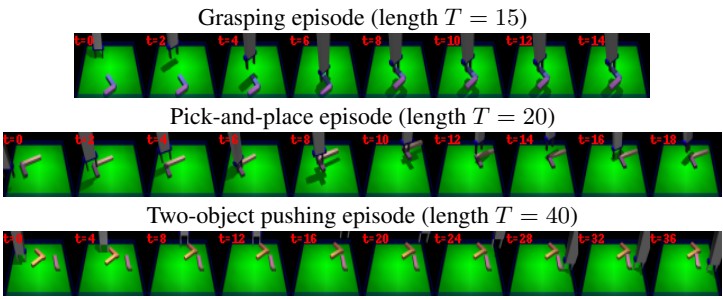

Pick-and-place episode (length $T = 20$)

Two-object pushing episode (length $T = 40$)

Figure 3: (Best seen in pdf) One sample episode each for grasping, pick-and-place, and pushing. Time overlaid on each frame.

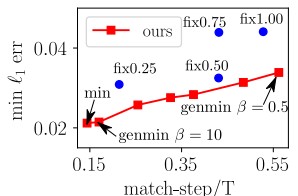

Figure 4: Forward prediction $\ell_1$ error for grasping. TAP methods (red) perform better than fixed-time predictors over all time steps.

## 4 EXPERIMENTS

We have proposed a time-agnostic prediction (TAP) paradigm that is different from the fixed-time paradigm followed in prior prediction work. In our experiments, we focus on comparing TAP against a representative fixed-time prediction model, keeping network architectures fixed. We use three simulated robot manipulation settings: object grasping (50k episodes), pick-and-place (75k episodes), and multi-object pushing (55k episodes). Example episodes from each task are shown in Fig 3 (videos in Supp). 5% of the data is set aside for testing. We use $64 \times 64$ images.

For grasping (15 frames per episode), the arm moves to a single object on a table, selects a grasp, and lifts it vertically. For pick-and-place (20 frames), the arm additionally places the lifted object at a new position before performing a random walk. For pushing (40 frames), two objects are initialized at random locations and pushed to random final positions. Object shapes and colors in all three settings are randomly generated. Fig 3 shows example episodes.

**Forward prediction.** First, we evaluate our approach for forward prediction on grasping. The first frame ("start") is provided as input. We train fixed-time baselines (architecture same as ours, using $\ell_1$ and GAN losses same as MIN and GENMIN) that target predictions at exactly $0.25, 0.50, 0.75, 1.0$ fraction of the episode length (FIX0.25,..., FIX1.00). MIN and GENMIN are TAP with/without the generalized minimum of Sec 3.2. For GENMIN, we evaluate different choices of the time preference vector $w(t)$ (Sec 3.2). We set $w(t) = \beta + t/15$, so that our preference increases linearly from $\beta$ to $\beta + 1$. Since $w(t)$ applies multiplicatively, low $\beta$ corresponds to high disparity in preferences ($\beta = \infty$ reduces to MIN, i.e., no time preference). GENMIN2 is our approach with $\beta = 2$ and so on.

Fig 5 shows example predictions from all methods for the grasping task. In terms of visual quality of predictions and finding a semantically coherent bottleneck, GENMIN2, GENMIN4, and GENMIN7 perform best—they reliably produce a grasp on the object while it is still on the table. With little or no time preferences, MIN and GENMIN10 produce images very close to the start, while GENMIN0.5 places too high a value on predictions farther away, and produces blurry images of the object after lifting.

Quantitatively, for each method, we report the min and arg min index of the $\ell_1$ distance to all frames in the video, as "min $\ell_1$ err" and "match-step" ("which future ground truth frame is the prediction closest to?"). Fig 4 shows a scatter plot, where each dot or square is one model. TAP (various models with varying $\beta$) produces an even larger variation in stepsizes than fixed-time methods explicitly targeting the entire video (FIX0.75 and FIX1.0 fall short of producing predictions at 0.75 and 1.0 fraction of the episode length). TAP also produces higher quality predictions (lower error) over that entire range. From these quantitative and qualitative results, we see that TAP not only successfully encourages semantically coherent bottleneck predictions, it also produces higher quality predictions than fixed-time prediction over a range of time offsets.

**Intermediate frame prediction.** Next, we evaluate our approaches for bidirectionally conditioned prediction in all three settings. Initial and final frames are provided as input, and the method is trained to generate an intermediate frame. The FIX baseline now targets the middle frame. As before, MIN and GENMIN are our TAP models. The GENMIN time preference $w(t)$ is bell-shaped and varies from $2/3$ at the ends to 1 at the middle frame (see Appendix E).

Figs 6 and 9 show examples from the three settings. TAP approaches successfully discover interesting bottlenecks in each setting. For grasping (Fig 6 (left)), both MIN and GENMIN consistently produce

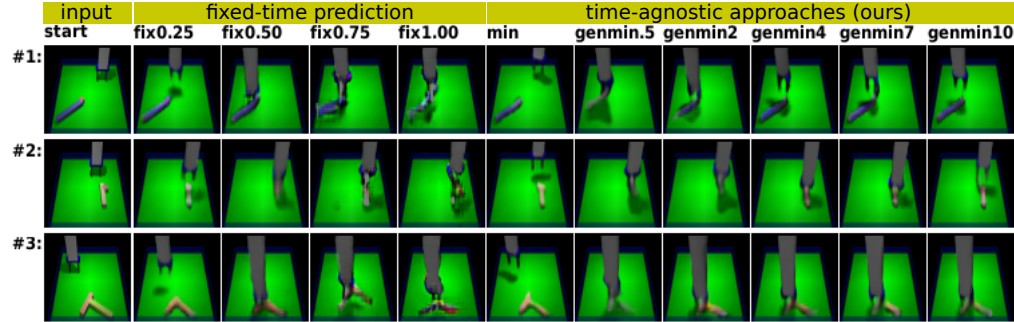

Figure 5: (Best seen in pdf) Forward prediction results on grasping comparing fixed-time predictors and our approach. Each row is a separate example. First column is the input. Thereafter, each column corresponds to the output of a different model per the column title. More in Appendix Fig 17.

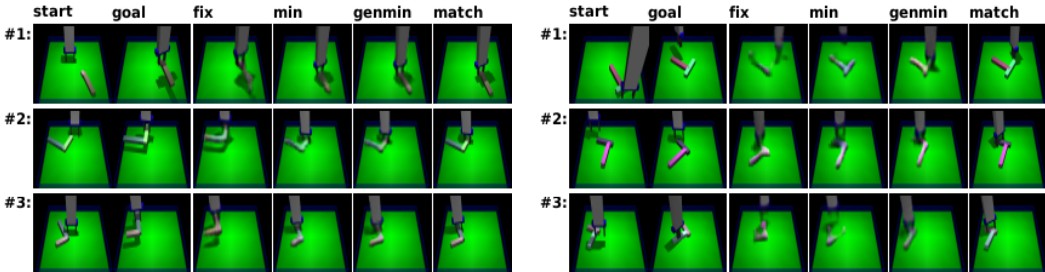

Figure 6: (Best seen in pdf) Bidirectional prediction results comparing fixed-time prediction and our approach. **(Left)** Grasping results. First two columns are inputs (start and goal). Thereafter, each column corresponds to the output of a different model per the column title. "match" is the ground truth image closest to the GENMIN prediction. More in Appendix Fig 13. **(Right)** Similar results for pick-and-place. More in Appendix Fig 14.

clear images of the arm at the point at which it picks up the object. Pick-and-place (Fig 6, right) is harder because it is more temporally extended, and the goal image does not specify how to grasp the object. FIX struggles to produce any coherent predictions, but GENMIN once again identifies bottlenecks reliably—in examples #3 and #1, it predicts the "pick" and the "place" respectively. For the pushing setting (Fig 9 (left)), GENMIN frequently produces images with one object moved and the other object fixed in place, which again is a semantically coherent bottleneck for this task. In row #1, it moves the correct object first to generate the subgoal, so that objects do not collide.

Table 1 shows quantitative results over the full test set. As in forward prediction, we report min $\ell_1$ error and the best-matching frame index ("match-step") for all methods. MIN and GENMIN consistently yield higher quality predictions (lower error) than FIX at similar times on average. As an example, GENMIN reduces FIX errors by 21%, 26.5%, and 53.2% on the three tasks—these are consistent and large gains that increase with increasing task complexity/duration. Additionally, while all foregoing results were reported without the CVAE approach of Sec 3.4, Table 1 shows results for GENMIN+VAE, and Fig 9 shows example predictions for pick-and-place. In our models, individual stochastic predictions from GENMIN+VAE produce higher $\ell_1$ errors than GENMIN. However, the CVAE helps capture meaningful sources of stochasticity at the bottlenecks—in Fig 9 (right), it produces different grasp configurations to pick up the object in each case. To measure this, we evaluate the best of 100 stochastic predictions from GENMIN+VAE in Table 1 (GENMIN+VAE BEST-OF-100). On pick-and-place

| Setting → Method ↓ | Grasping (T=15 steps) | | Pick-and-place (T=20 steps) | | Pushing (T=30 steps) | |
|---|---|---|---|---|---|---|
| | min $\ell_1$ err | match-step/T | min $\ell_1$ err | match-step/T | min $\ell_1$ err | match-step/T |
| fix | 0.0153 | 0.51±0.17 | 0.0366 | 0.53±0.24 | 0.0722 | 0.36±0.19 |
| MIN (ours) | **0.0104** | 0.49±0.18 | 0.0256 | 0.41±0.30 | 0.0365 | 0.35±0.22 |
| GENMIN (ours) | 0.0121 | 0.45±0.16 | 0.0269 | 0.42±0.23 | **0.0338** | 0.36±0.19 |
| GENMIN W/O GAN (ours) | 0.0117 | 0.45±0.16 | **0.0235** | 0.46±0.25 | 0.0411 | 0.37±0.19 |
| GENMIN + VAE (ours) | 0.0156 | 0.47±0.18 | 0.0432 | 0.31±0.24 | 0.0447 | 0.37±0.21 |
| GENMIN + VAE BEST-OF-100 (ours) | 0.0121 | - | **0.0196** | - | **0.0236** | - |

Table 1: Bidirectional frame prediction performance on: grasping, pick-and-place, and two-object pushing. Lower min $\ell_1$ err is better. match-step denotes which times are being predicted, out of $T$ steps. It is clear that TAP methods make better predictions than fixed-time prediction at the same time offsets.

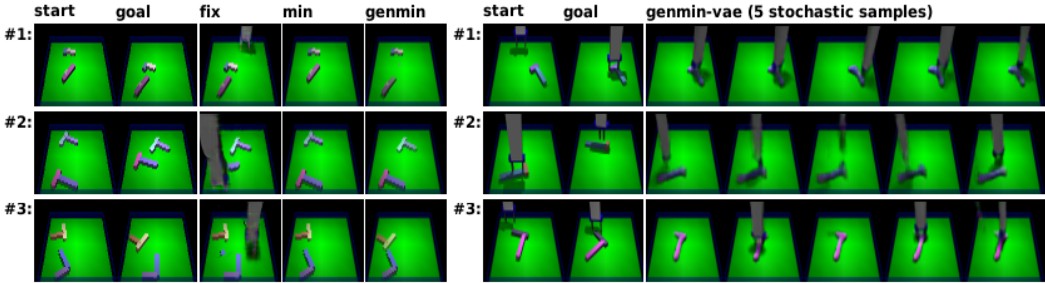

Figure 9: (Best seen in pdf) **(Left)** Bidirectional prediction results on two-object pushing. More in Appendix Fig 15. **(Right)** When used with a VAE (Sec 3.4), our approach captures residual stochasticity at the bottleneck. In these results from the pick-and-place task, GENMIN+VAE produces images that are all of the arm in contact with the object on the table, but at different points on the object, and with different arm/gripper poses.

and pushing, the best VAE results are significantly better than any of the deterministic methods. Table 1 also shows results for our method without the GAN (GENMIN W/O GAN)—while its $\ell_1$ errors are comparable, we observed a drop in visual quality.

As indicated in Sec 3.1 and Eq 3, TAP may also be applied recursively. Fig 7 compares consecutive subgoals for the pick-and-place task produced by recursive TAP versus a recursive fixed-time model. Recursion level $r = 1$ refers to the first subgoal, and $r = 2$ refers to the subgoal generated when the first subgoal is provided as the goal input (start input is unchanged). In example #1, FIX struggles while "ours" identifies the "place" bottleneck at $r = 1$, and subsequently the "pick" bottleneck at $r = 2$.

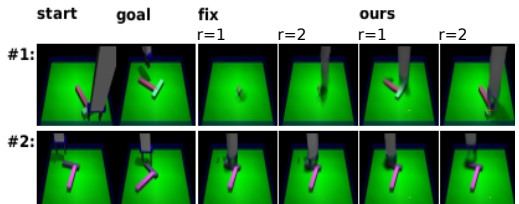

Figure 7: Recursive bidirectional prediction on pick-and-place. $r = 2$ is earlier in time than $r = 2$.

Finally, we test on "BAIR pushing" (Ebert et al., 2017), a real-world dataset that is commonly used in visual prediction tasks. The data consists of 30-frame clips of random motions of a Sawyer arm tabletop. While this dataset does not have natural bottlenecks like in grasping, TAP (min $\ell_1$ error 0.046 at match-step 15.42) still performs better than FIX (0.059 at 15.29). Qualitatively, as Fig 8 shows, even though BAIR pushing contains incoherent random arm motions, TAP consistently produces predictions that plausibly lie on the path from start to goal image. In example #1, given a start and goal image with one object displaced, "ours" correctly moves the arm to the object before displacement, whereas FIX struggles.

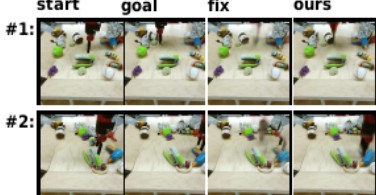

Figure 8: Bidirectional prediction results on BAIR pushing data. The first two columns are the inputs, and the next two correspond to FIX and GENMIN.

**Bottleneck discovery frequency.** We have thus far relied on qualitative results to assess how often our approach finds coherent bottlenecks. For pushing, we test bottleneck discovery frequency more quantitatively. We make the reasonable assumption that bottlenecks in 2-object pushing correspond to states where one object is pushed and the other is in place. Our metric exploits knowledge of true object positions at start and goal states. First, for this evaluation, we restrict both GENMIN and FIX to synthesize predictions purely by warping and masking inputs. Thus, we can track where the pixels at ground truth object locations in start and goal images end up in the prediction, i.e., where did each object move? We then compute a score that may be thresholded to detect when only

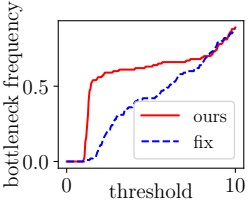

Figure 10: Bottleneck frequency vs. score threshold

one object moves (details in Appendix F). As an intuitive example, suppose that the two objects are displaced by 10 cm and 15 cm respectively between start and goal frames. Suppose further that our predictor predicts the start frame as its output. Then its distance score would be computed to be 10 cm. For all values of threshold below 10 cm, this would be counted as a bottleneck discovery failure.

As Fig 10 shows, GENMIN predicts bottlenecks much more frequently ($\sim$60% of the time) than FIX. As hypothesized, our time-agnostic approach does indeed identify and latch on to low-uncertainty states to improve its predictions.

**Hierarchical planning evaluation.** Finally, we discuss experiments directly evaluating our intermediate predictions as visual subgoals for hierarchical planning for pushing tasks. A forward Visual MPC planner (Ebert et al., 2017) accepts the subgoal object positions (computed as above for evaluating bottlenecks). Start and goal object positions are also known. Internally, Visual MPC makes action-conditioned fixed-time forward predictions of future object positions to find an action sequence that reaches the subgoal object

|  | 2-object | 3-object |
|---|---|---|
| direct | 12.9±0.6 | 15.8±0.6 |
| FIX | 12.5±0.5 | 17.6±0.6 |
| GENMIN | **11.9±0.6** | **12.9±0.7** |

Table 2: Multi-object pushing errors (in cm).

positions, with a planning horizon of 15. Additional implementation details are in Appendix G.

Given start and goal images, our model produces a subgoal. Visual MPC plans towards this subgoal for half the episode length, then switches to the final goal. We compare this scheme against (i) "direct": planning directly towards the final goal for the entire episode length, and (ii) FIX: subgoals from a center-frame predictor. The error measure is the mean of object distances to goal states (lower is better). As an upper bound, single-object pushing with the planner yields $\sim$5cm error. Results for two-object and three-object pushing are shown in Table 2. GENMIN does best on both, but especially on the more complex three-object task. Since Visual MPC has thus far been demonstrated to work only on pushing tasks, our hierarchical planning evaluation is also limited to this task. Going forward, we plan to adapt Visual MPC to allow testing TAP on more complex temporally extended tasks like block-stacking, where direct planning breaks down and subgoals offer greater value.

## 5 CONCLUSIONS

The standard paradigm for prediction tasks demands that a predictor not only make good predictions, but that it make them on a set schedule. We have argued for redefining the prediction task so that the predictor need only care that its prediction occur at *some* time, rather than that it occur at a specific scheduled time. We define this time-agnostic prediction task and propose novel technical approaches to solve it, that require relatively small changes to standard prediction methods. Our results show that reframing prediction objectives in this way yields higher quality predictions that are also semantically coherent—unattached to a rigid schedule of regularly specified timestamps, model predictions instead naturally attach to specific semantic "bottleneck" events, like a grasp. In our preliminary experiments with a hierarchical visual planner, our results suggest that such predictions could serve as useful subgoals for complex tasks.

In future work, we would like to address some limitations of our TAP formulation, of which we will mention two here. First, TAP currently benefits not only from selecting which times to predict, but also from not having to provide timestamps attached to its predictions. We would like to study: could we retain the benefits of time-agnostic prediction while also providing timestamps for when each predicted state will occur? Second, our current TAP formulation may not generalize to prediction problems in all settings of interest. As an example, for videos of juggling or waving, which involve repeated frames, TAP might collapse to predicting the input state repeatedly. We would like to investigate more general TAP formulations: for example, rather than choosing $\mathcal{E}_t^J$ in Eq 5 to encourage predicting farther away times, we could conceivably penalize predictions that look too similar to the input context frames. More broadly, we believe that our results thus far hold great promise for many applications of prediction including hierarchical planning and model-based reinforcement learning, and we hope to build further on these results.

**Acknowledgements.** We thank Alex Lee and Chelsea Finn for helpful discussions and Sudeep Dasari for help with the simulation framework and for generating the simulated data used in this work. We thank Kate Rakelly, Kyle Hsu, and Allan Jabri for feedback on early drafts. This work was supported by Berkeley DeepDrive, NSF IIS-1614653, NSF IIS-1633310, and an Office of Naval Research Young Investigator Program award. The NVIDIA DGX-1 used for this research was donated by the NVIDIA Corporation.

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

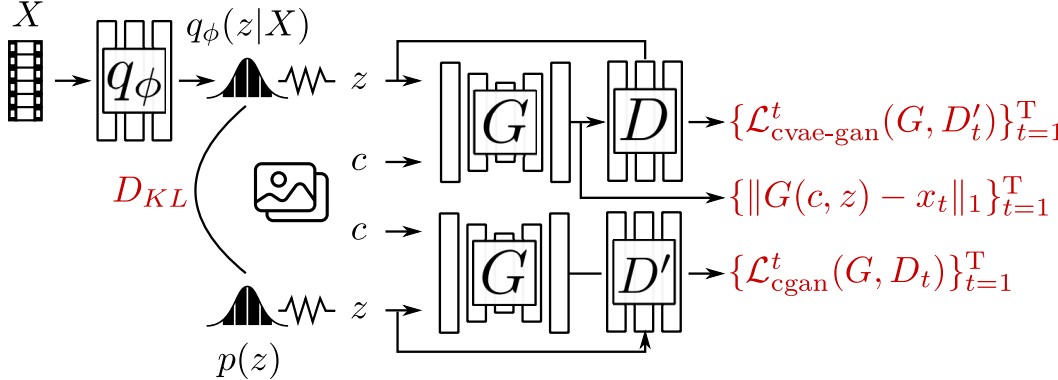

Figure 11: Training time network schematic. At test time, only the predictor $G$ is used, and $z \sim \mathcal{N}(0, \mathcal{I})$. Loss terms (as used in Eq 10) are in red.

In these appendices, we provide details omitted in the main text for space. Note that more supplementary material, such as video examples, is hosted at: `https://sites.google.com/view/ta-pred`

## A  LABEL SMOOTHING FOR TIME-AGNOSTIC CONDITIONAL GANS

In Eq 7, we defined a time-agnostic CGAN loss that trained T discriminators, one corresponding to each target time $t \in T$. In Eq 7, the only positives for the $t$-th discriminator $D_t$ were ground truth frames that occured precisely at time $t$ relative to the input $c$, i.e., discriminators are trained so that $D_t(c, x_t)$ would be 1, and $D_t(c, x_{t' \neq t})$ and $D_t(c, G(c))$ would both be 0. However, in practice, we use the following slightly different loss:

$$G^* = \arg\min_{G} \min_{t \in T} \max_{D_t} \mathcal{L}_{\text{cgan}}^t(G, D_t),$$

$$\mathcal{L}_{\text{cgan}}^t(G, D_t) = \log D_t(c, x_t) + \log(1 - D_t(c, G(c))) +$$

$$\sum_{t' \neq t} [l_{t,t'} \log D_t(c, x_{t'}) + (1 - l_{t,t'}) \log(1 - D_t(c, x_{t'}))], \quad (11)$$

where we set $l_{t,t'} = \max(0, 1 - \alpha|t - t'|)$ with $\alpha = 0.25$. The first two terms are the same as Eq 7 — for the $t$-th discriminator, the $t$-th frame provides a positive, and the generated frame provides a negative instance. The third term defines the loss for frames $x_{t' \neq t}$: frames close to time $t$ are partial positives ($0 < l_{t,t'} < 1$), and others are negatives ($l_{t,t'} = 0$). This label smoothing stabilizes training and ensures enough positives for each discriminator $D_t$.

## B  TIME-AGNOSTIC CONDITIONAL VAES

Here, we discuss one alternative to the TAP CVAE formulation of Eq 9 in Sec 3.4. Rather than restricting the minimum-over-time to be over the log-likelihood term alone, why not take the minimum-over-time over the whole loss? In other words, the inference network would still have to only see one frame as input, and this version of the TAP CVAE loss would be:

$$\mathcal{L}_{\text{cvae}}(G, \phi) = \min_{t \in T} \left[ D_{KL}(q_\phi(z|x_t), p(z)) - \mathbb{E}_{z \sim q_\phi(z|x_t)} \ln p_G(x_t|c, z) \right]. \quad (12)$$

Unfortunately, this would be computationally very expensive to train. Both the inference network $q_\phi$ and the generator $G$ would have to process $|T|$ frames. In comparison, the formulation of Eq 9 allows $q_\phi$ and $G$ to process just one frame, and only error computation must be done T times, once with each target frame $x_t$ for $t \in T$.

## C  ARCHITECTURE AND TRAINING DETAILS

**Predictor architecture.**  As indicated in Fig 11, the predictor $G$ has an encoder-decoder architecture. When our approach is used together with the conditional VAE (Sec 3.4), the conditional latent $z$

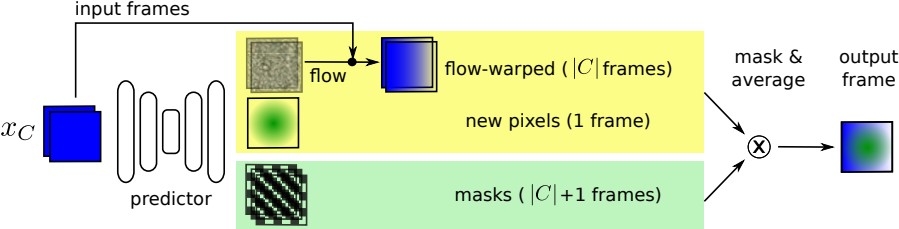

Figure 12: The predictor produces an appearance flowfield which is used to produce flow-warped input frames, which together with a frame of new pixels are masked and averaged to produce the output frame.

is appended at the bottleneck between the encoder and the decoder. The encoder produces a 128-dimensional code, and the VAE latent code $z$ is 32-dimensional, so the overall size of the input to the decoder is 160 when the VAE is used.

The **encoder** inside the predictor uses a series of 4x4convolution-batchnorm-LeakyReLU(0.2) blocks to reduce the 64x64 input image as 3x64x64→32x32x32→64x16x16→128x8x8→256x4x4. For bidirectional prediction, this is repeated for both images and concatenated to produce a 512x4x4 feature map. Finally, a small convolution-ReLU-convolution head reduces this to a 128-dimensional vector. Except for this last head subnetwork, this architecture is identical to the DCGAN discriminator (Radford et al., 2015).

The **decoder** inside the predictor uses a DCGAN-based architecture (Radford et al., 2015). The input vector is reshaped by a transposed convolution into 256 4×4 maps. This is subsequently processed a series of bilinear upsampling-5x5convolution-batchnorm-ReLU blocks as 256x4x4→128x8x8→64x32x32→3x64x64. For the last block, a Tanh activation is used in place of ReLU to keep outputs in [-1,1]. Compared to (Radford et al., 2015), the main difference is that transposed convolutions are replaced by upsampling-5x5 convolution blocks, which aids in faster learning with fewer image artifacts (Odena et al., 2016).

As shown in Fig 12, our predictor produces three sets of outputs: (a) one frame of new pixels, (b) "appearance flow" (Zhou et al., 2016) maps that warp the $|C|$ input image(s), and (c) $|C| + 1$ masks (summing to 1 at each pixel) that combine the warped input images and the new pixels frame to produce the final output. To produce these three outputs, we use three decoders that all have the same architecture as above, except that the final output is shaped appropriately—the appearance flow decoder produces $|C|$x64x64 flowfields, and the masks decoder produces $(|C| + 1)$x64x64 masks.

**Discriminator and Recognition Network** The VAE recognition network and the discriminator both use similar architectures to the predictor encoder above. Only the input layer and the output layer are changed as follows: (a) The discriminator accepts $|C| + 1$ images as input and the recognition network accepts $|C| + |T|$ images (the full video) as input. (b) The discriminator head produces $|T|$ logits (one corresponding to each target time), and the recognition network produces a 32-dimensional conditional latent.

**Training.** We found it beneficial to initialize the decoder by training it first as an unconditional frame generator on frames from the training videos. For this pretraining, we use learning rate 0.0001 for 10 epochs with batch size 64 and Adam optimizer. Thereafter, for training, we use learning rate 0.0001 for 200 epochs with batch size 64 and Adam optimizer.

# D DATA GENERATION

To generate the data, we use a cross-entropy method (CEM)-based planner (Kroese et al., 2013; Ebert et al., 2017) in the MuJoCo (Todorov et al., 2012) environment with access to the physics engine, which produces non-deterministic trajectories. The planner plans towards randomly provided goals, but we use both successful and failed trajectories. Sample videos of episodes are shown at: `https://sites.google.com/view/ta-pred`

## E  GENERALIZED MINIMUM WEIGHTS FOR INTERMEDIATE PREDICTION

In Sec 4, we briefly mentioned that the time preference $w(t)$ for the generalized minimum loss during intermediate frame prediction was set to 2/3 at the ends to 1 at the middle frame. In our experiments, we set these weights heuristically as follows: $w(t)$ rises linearly from $\kappa = 0.66$ at $t = 1$ to 1.0 at the $t = 5$, then stay at 1.0 for $T - 10$ frames. Then, starting from the $(T - 5)$-th frame, it would drop linearly to $\kappa$ once more at $t = T$. The only hyperparameter we tuned was $\kappa$ (search over 2/3 and 1/3).

## F  BOTTLENECK DISCOVERY FREQUENCY SCORE

In Sec 4, we mentioned a bottleneck discovery frequency measure in the paragraph titled "Bottleneck discovery frequency metric." We now provide further details.

Our proposed metric for two-object pushing quantifies how reliably the network is able to generate a bottleneck state with one object being moved and the other being at its original position. The reason that this state is of interest is that in two-object pushing, this may be reasonably assumed to be the natural bottleneck, so we call this metric the bottleneck frequency. Even without this assumption though, the metric quantifies the ability of our approach to generate predictions attached to this consistent semantically coherent bidirectional state.

To measure bottleneck frequency, we use a technique similar to the approach proposed in (Ebert et al., 2017) for planning with a visual predictor. First, we train versions of "genmin" and "fix" that synthesize predictions purely by appearance-flow-warping and masking inputs (as shown in the scheme of Fig 12, but without pixel generation). Next, recall that we have access to the starting and goal positions of objects since our dataset is synthetically generated. Thus, we can exploit this and track where the pixels at ground truth object locations in start and goal images end up in the prediction, i.e., where did each object move? This works as follows: we take the appearance flow transformations and masks generated by the model internally (for application to input images to generate prediction) and apply them instead to one-hot object location maps—these maps have value one at the ground truth origin of the the objects and zero elsewhere. The output of this operation is a probability map for each object indicating where it is located in the predictor's bidirectional prediction output.

To compute the score, we then calculate the expected distance in pixels between the predicted positions of each object and the bottleneck state. There are actually two possible candidates for this bottleneck state: object 1 is moved first, or object 2 is moved first. We compute the expected distances to both these bottleneck candidates and take the lower distance to be the score. This score does not evaluate whether the semantically correct bottleneck was predicted (in cases where one object must always be moved first to avoid collision, such as Fig 9 (left) example #1).

The lower this distance score, the higher the likelihood that the predicted output is actually a bottleneck ("bottleneck frequency"). Fig 10 shows what happens when the threshold over the score is varied, for our approach and the fixed-time baseline. Higher bottleneck frequency at lower threshold is desirable. As the figure shows, at a low threshold distance score ($\approx 2$ pixels), our approach gets to about 58% bottleneck frequency while the fixed-time predictor gets about 0% frequency at this threshold. This verifies that our approach produces predictions attached to semantically coherent low-uncertainty bottleneck events.

## G  HIERARCHICAL PLANNING EVALUTION METHOD

In Sec 4, we described a hierarchical planning approach using our predictions in the paragraph titled "Hierarchical planning evaluation." We describe this method in more detail here.

We test the usefulness of the generated predictions as subgoals for planning multi-object pushing. Given a start and goal image, we produce an bidirectional prediction using our time-agnostic predictor and feed it to a low-level planner that plans towards this prediction as its subgoal. For the low-level planner we use the visual model-predictive control algorithm ("Visual MPC") (Ebert et al., 2017) which internally uses a fixed-time forward prediction model and sampling-based planning to solve short-term and medium duration tasks. A more detailed description of this process follows.

Visual MPC requires start and goal locations of objects to plan. When used in conjunction with our method, we first compute the locations of objects at the bidirectional predictions and feed this in place of the final goal object locations, so that Visual MPC may plan to first reach the bidirectional prediction as a subgoal/stepping stone towards the final goal. Once the subgoal is reached, Visual MPC is fed the final goal. To compute subgoal object locations, we use the same technique as in Appendix F above—one-hot maps of object locations are transformed by the appearance flow maps and masks computed internally by our predictor. This produces a probability map $p_g$ over object locations at the prediction, which is passed to Visual MPC.

Internally, the Visual MPC planner makes fixed-time forward predictions for the object locations starting from the initial distribution $p_s$ given a randomly sampled action sequence of length $h = 15$ (the "planning horizon"). Out of all the action sequences, it selects the sequence that brings the distribution closest to $p_g$ within $h = 15$ steps ($h$ is the "planning horizon"). In practice, we use 200 random action sequences and perform three iterations of CEM search (Kroese et al., 2013) to find the the best sequence. Once an action sequence is selected, the first action in the sequence is executed, and the initial object distribution is updated. New candidate action sequences are now generated starting from this updated object distribution and the process repeats.

In our experiments, we use a time budget $B$ of 40 steps for 2-object pushing, and 75 steps for 3-object pushing. In both cases, we feed Visual MPC the subgoal (bidirectional prediction object locations) for the first $B/2$ timesteps, and then feed the final goal location for the last $B/2$ steps. The "direct" planning baseline instead receives the final goal location for all $B$ steps. As shown by the results in the paper, our subgoal predictions aid in performing these multi-stage pushing tasks better than with the direct planning baseline.

## H  SUPPLEMENTARY PREDICTION EXAMPLES

We now show more prediction results from various settings in Figs 13, 14, 15, 16, and 17, to supplement those in Sec 4.

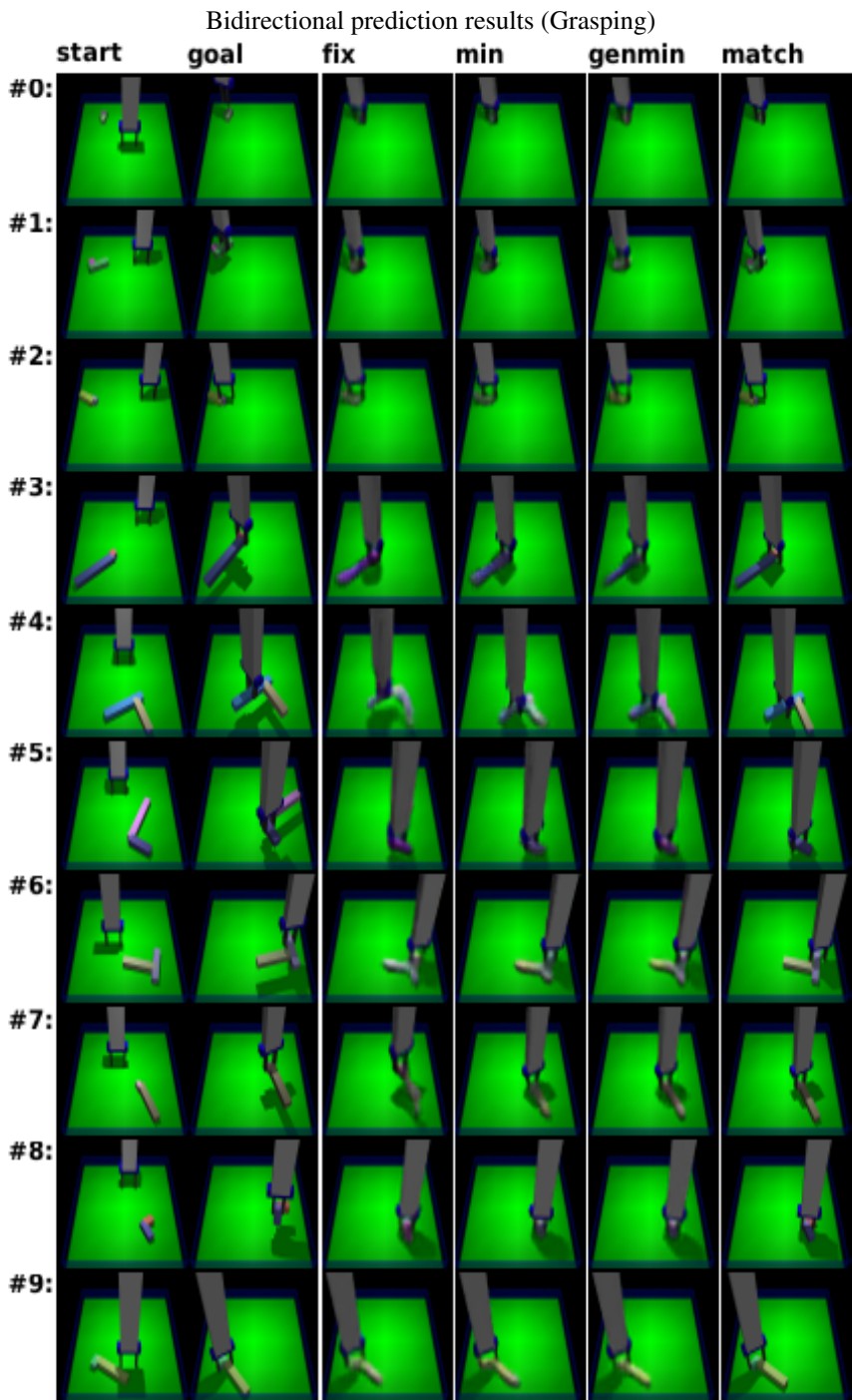

Figure 13: (Same format as Fig 6). Supplementary bidirectional prediction results comparing fixed-time prediction and our approach on grasping. First two columns are inputs (start and goal). Thereafter, each column corresponds to the output of a different model per the column title. "match" is the ground truth image closest to the GENMIN prediction.

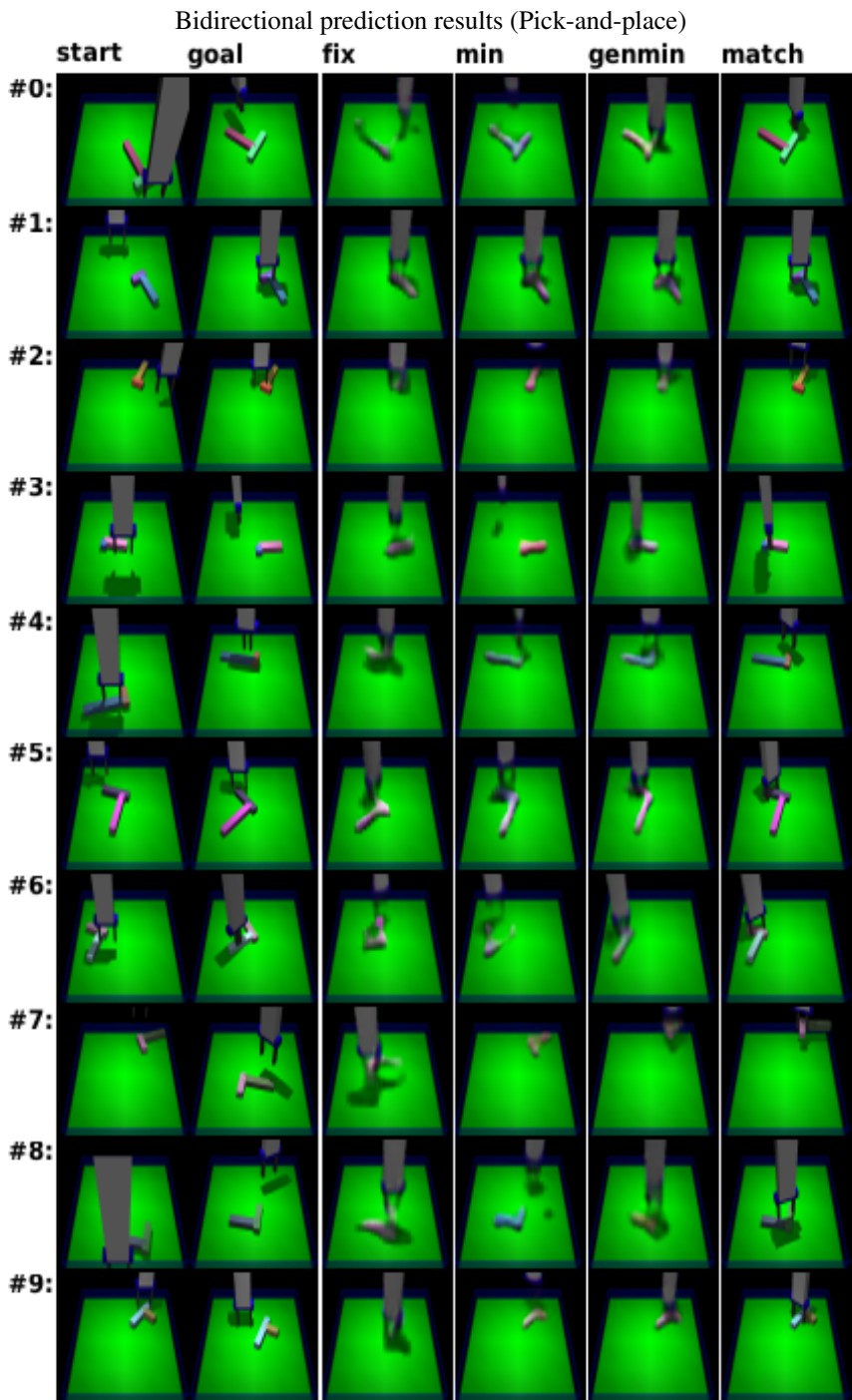

Figure 14: (Same format as Fig 6). Supplementary bidirectional prediction results comparing fixed-time prediction and our approach on pick-and-place. First two columns are inputs (start and goal). Thereafter, each column corresponds to the output of a different model per the column title. "match" is the ground truth image closest to the GENMIN prediction.

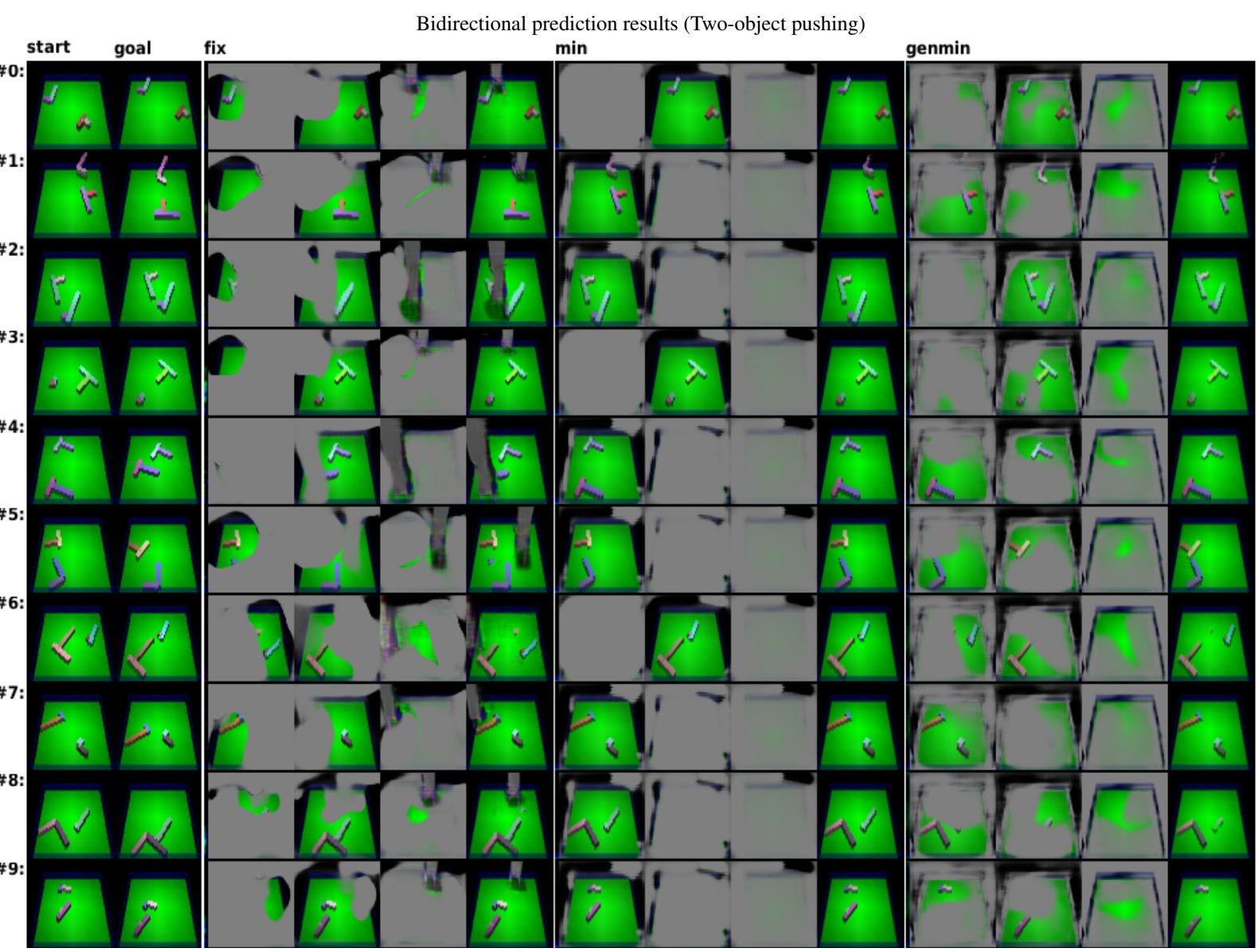

Figure 15: Supplementary bidirectional prediction results on pushing. Four columns for each method correspond to masked warped start frame, masked warped goal frame, masked new pixels, and final output respectively. See Appendix C and Fig 12 for the synthesis scheme with masks and warps.

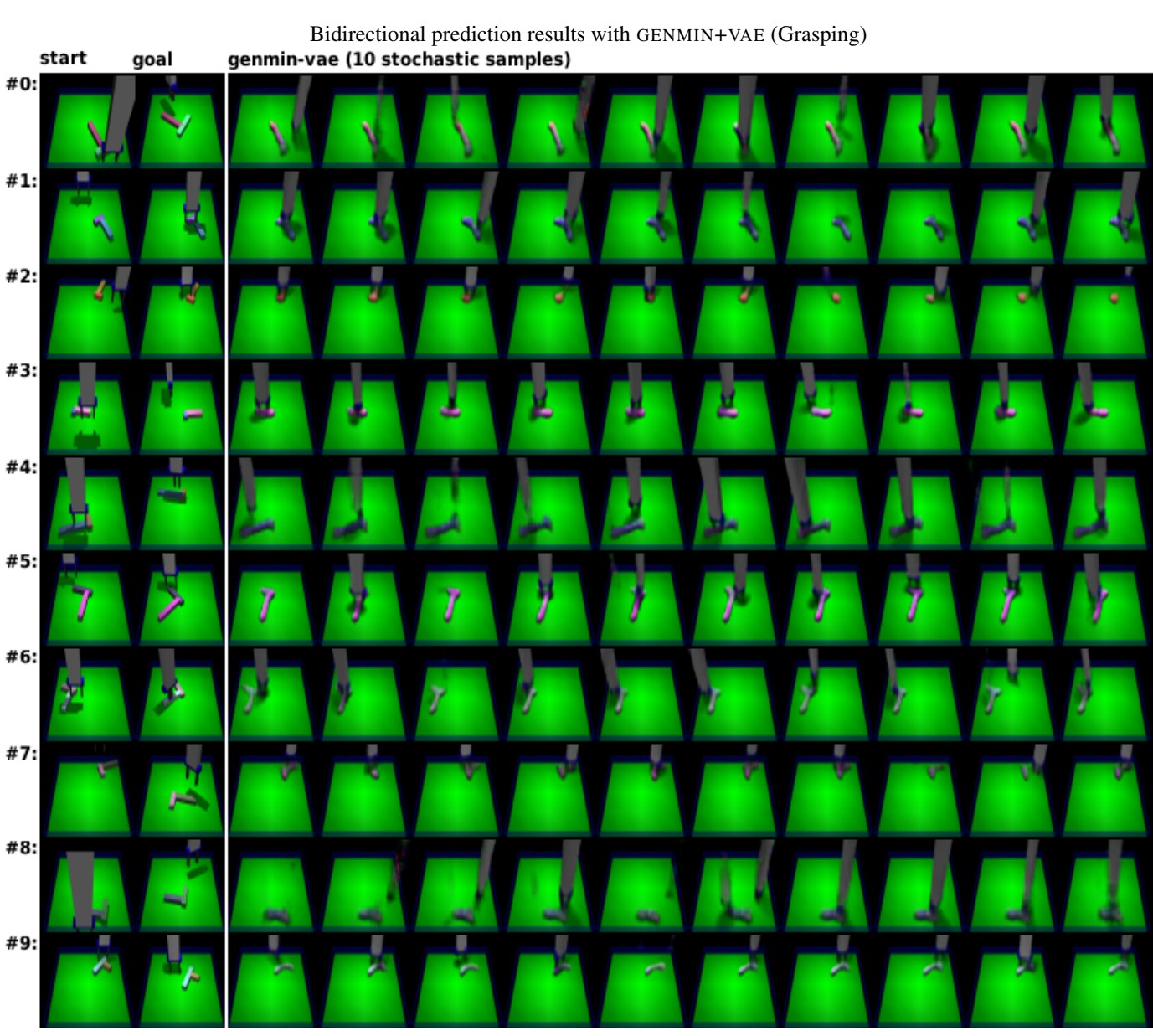

Figure 16: Supplementary bidirectional prediction examples for pick-and-place with GENMIN+VAE. Same format as Fig 9. Each row is a separate example. First column is the input. GENMIN+VAE captures residual stochasticity at the bottleneck. GENMIN+VAE produces images that are most all of the arm in contact with the object on the table, but at different points on the object, and with different arm/gripper poses.

Forward prediction results (Grasping)

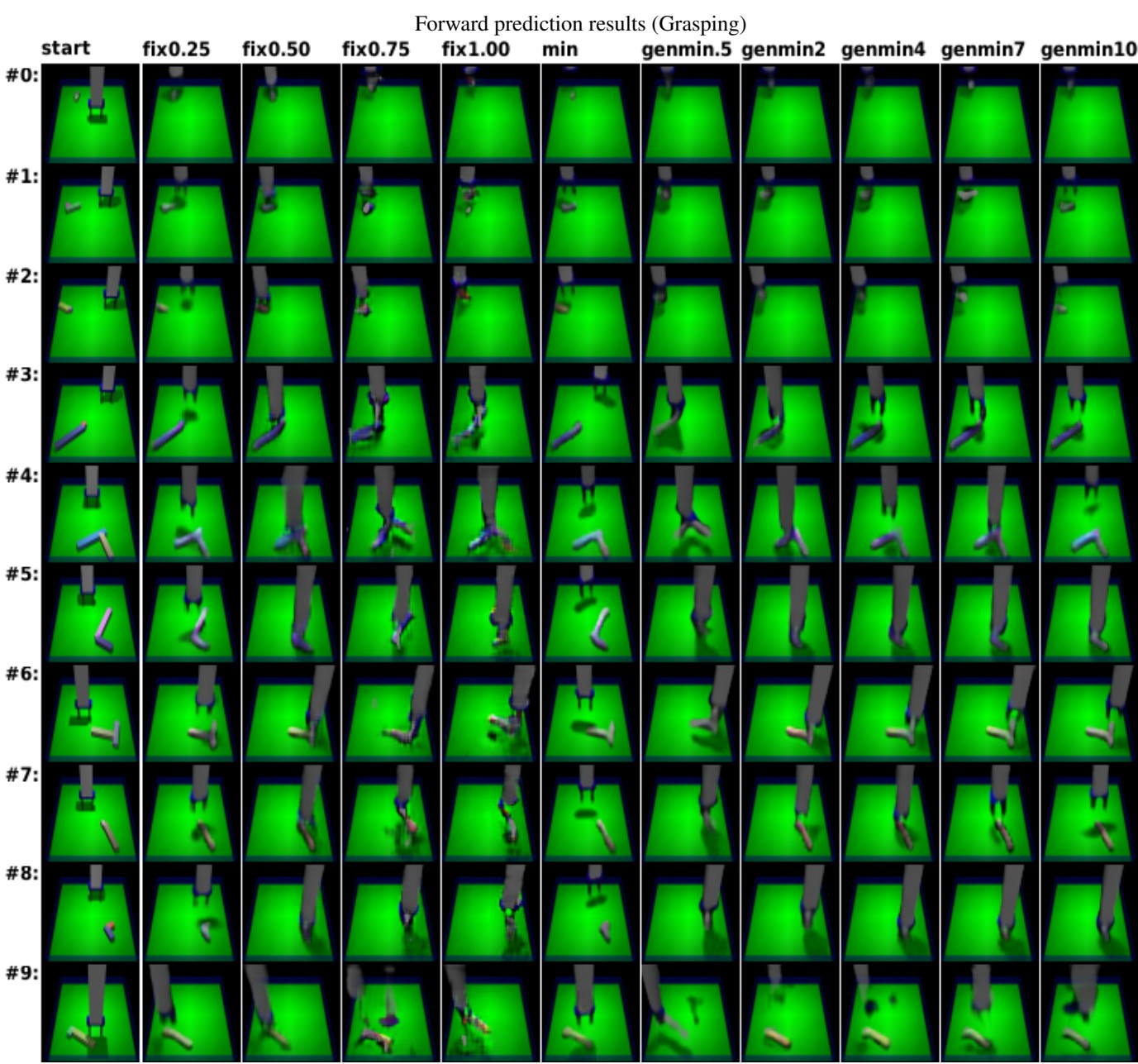

Figure 17: Supplementary forward prediction examples for grasping, comparing fixed-time predictors and our approach. Same format as Fig 5. Each row is a separate example. First column is the input. Thereafter, each column corresponds to the output of a different model per the column title.

