# OpenReview forum: "Time-Agnostic Prediction: Predicting Predictable Video Frames"
_ICLR.cc/2019/Conference_

### Official Review · AnonReviewer3 · 2018-11-02
**Good novel contribution**

**Rating:** 7
**Confidence:** 5

**Review:**

Summary:
The paper reformulates the task of video prediction/interpolation so that a predictor is not forced to generate frames at fixed time intervals, but instead it is trained to generate frames that happen at any point in the future. The motivation for such approach is that there might be future states that are highly uncertain – and thus, difficult to predict – that might not be useful for other tasks involving video prediction such as planning. The authors derive different loss functions for such Time-Agnostic Prediction (TAP), including extensions to the Variational AutoEncoders (VAE) and Generative Adversarial Networks (GAN) frameworks, and conduct experiments that suggest that the frames predicted by TAP models correspond to ‘subgoal’ states useful for planning.

Strenghts:
[+] The idea of TAP is novel and intuitively makes sense.
It is clear that there are frames in video prediction that might not be interesting/useful yet are difficult to predict, TAP allows to skip such frames.
[+] The formulation of the TAP losses is clear and well justified.
The authors do a good job at showing a first version of a TAP loss, generalizing it to express preferences, and its extension to VAE and GAN models, showing that

Weaknesses:
[-] The claim that the model discovers meaningful planning subgoals might be overstated.
The hierarchical planning evaluation experiment seems like it would clearly favor TAP compared to a fixed model (why would the middle prediction in time of the fixed model correspond to reasonable planning goals?). Furthermore, for certain tasks and environments it seems like the uncertain frames might be the ones that correspond to important subgoals. For example, for the BAIR Push Dataset, usually the harder frames to predict are the arm-object interactions, which probably would correspond to the relevant subgoals.

Overall I believe that the idea in this paper is a meaningful novel contribution. The paper is well-written and the experiments support the fact that TAP might be a better choice for training frame predictors for certain tasks.

---

> ### Author Response · Authors · 2018-11-15
> **(R3 review response) Generality of planning subgoals provided by Time-Agnostic Prediction**
>
> Thank you for your feedback and careful observations.
>
> -------------------------------------------------------------------------
> * “The hierarchical planning evaluation experiment seems like it would clearly favor TAP compared to a fixed model (why would the middle prediction in time of the fixed model correspond to reasonable planning goals?).”
>
> Our hierarchical planning experiment is designed to evaluate the utility of a subgoal as follows: a planner spends half its time budget trying to move towards the subgoal, and the remaining half moving towards the end goal. Good subgoals are those that lead to better goal-reaching performance. Since the planner spends exactly half its time planning towards the subgoal, the middle frame is the most obvious choice for a subgoal---this is why FIX targets the middle frame. We will make this clearer in the text.
>
> However, if the above argument is not convincing, please let us know if the following experiment would help alleviate your concern. Rather than *predict* the middle frame, we could use the ground truth middle frame as a subgoal---let us call this baseline ”GT-MIDDLE”.  More specifically, we would sample a trajectory and the task would be to get from its start state to its end state, using the true middle image in that trajectory as a subgoal. This would answer the question: “if we were *perfectly* able to predict the middle frame, would that serve as a useful subgoal for planning?” If GT-MIDDLE is shown to work well, then we could conclude that predicting the middle frame would therefore provide a reasonable baseline.
>
> Please let us know if you believe that such an experiment would be informative and help to address your concern. Otherwise, do you perhaps have an alternative suggestion for a subgoal generation baseline?
>
> -------------------------------------------------------------------------
> * “Furthermore, for certain tasks and environments it seems like the uncertain frames might be the ones that correspond to important subgoals. For example, for the BAIR Push Dataset, usually the harder frames to predict are the arm-object interactions, which probably would correspond to the relevant subgoals.”
>
> As shown in the paper, in all our experiments, the low-uncertainty predictions from TAP have corresponded to semantically coherent task decompositions and intuitive subgoals. We believe there are good reasons to expect this to hold generally. Suppose that we had a training dataset that consisted of all possible trajectories between every pair of start and goal states. For example, in the maze in Fig 1, where start and goal states are fixed, suppose our training dataset included *every* possible successful trajectory. Then the easiest-to-predict frames would correspond to frames that occur in *every* possible trajectory between start and goal, which are intuitively good subgoals to aim for. In the maze of Fig 1, these would be the asterisks. Note that other positions in the maze (which would be more uncertain because they do not occur in all trajectories) could also be valid subgoals. But while the asterisks are guaranteed to lie on the shortest path from start to goal, more uncertain subgoals could lead to detours from this shortest path.
>
> More specifically for the BAIR pushing case in your comment, some example BAIR pushing videos are shown at https://sites.google.com/view/ta-pred/home#h.p_MyPmVZLyHypx. They consist of random arm motions, so that the arm is never continuously in contact with objects throughout the video. When an object is displaced in a video, our method usually produces images of the arm initiating or ending contact with that object, as shown in Fig 8. These correspond to low-uncertainty bottlenecks---the arm *must* have come into contact with that object, no matter how precisely the pushing motion occurred. We believe these states are also the most relevant subgoals in these cases, since they represent states that any successful trajectory has to pass through, as argued above.
>
> It is possible to imagine other more difficult settings than BAIR pushing, where bottlenecks would correspond to images of an arm dynamically pushing an object, which, we agree, is a harder prediction task. Since TAP is designed to select *relatively* easier frames, this would not affect it adversely; it would continue to predict the easiest among those difficult frames.
>
> --------------------------------------------------------
> Overall, we broadly agree it is difficult to rigorously claim that TAP *always* discovers meaningful subgoals, since there is no agreed-upon notion of what constitutes a good subgoal. In our responses above, we argue that TAP naturally targets one reasonable notion of a good subgoal --- a state that would occur in a large fraction of trajectories between start and goal states. We will qualify the subgoals claim more clearly in this way in the text if R3 agrees that this would be appropriate.

---

### Official Review · AnonReviewer1 · 2018-11-02
**Very interesting proposal and well-written method. Experiments section is though poorly structured**

**Rating:** 8
**Confidence:** 4

**Review:**

Revision
----------
Thanks for taking the comments on board. I like the paper, before and after, and so do the other reviewers. Some video results might prove more valuable to follow than the tiny figures in the paper and supplementary. Adding notes on limitations is helpful to understand future extensions.

-----------------------------------------------
Initial Feedback:
---------------------
This is a very exciting proposal that deviates from the typical assumption that all future frames can be predicted with the same certainty. Instead, motivated by the benefits of discovering bottlenecks for hierarchical RL, this work attempts to predict ‘predictable video frames’ – those that can be predicted with certainty, through minimising over all future frames (in forward prediction) or all the sequence (in bidirectional prediction). Additional, the paper tops this with a variational autoencoder to encode uncertainty, even within those predictable frames, as well as a GAN for pixel-level generation of future frames.

The first few pages of the paper are a joy to read and convincing by default without looking at experimental evidence. I do not work myself in video prediction, but having read in the area I believe the proposal is very novel and could make a significant shift in how prediction is currently perceived. It is a paper that is easy to recommend for publication based on the formulation novelty, topped with VAEs and GANs as/when needed.

Beyond the method’s explanation, I found the experiment section to be poorly structured. The figures are small and difficult to follow – looking at all the figures it felt that “more is actually less”. Many of the evidence required to understand the method are only included in the appendices. However, having spent the time to go back and forth, I believe the experiments to be scientifically sound and convincing.

I would have liked a discussion in the conclusion on the method’s limitation. This reviewer believes that the approach will struggle to deal with cyclic motions. In this case the discovered bottlenecks might not be the most useful to predict, as these will correspond to future frames (not nearby though) that are visually similar to the start (in forward) or to the start/end (in bidirectional) frames. An additional loss to reward difficult-to-predict frames (though certain compared to other times) might be an interesting additional to conquer more realistic (rather than synthetic) video sequences.

---

> ### Author Response · Authors · 2018-11-16
> **(R1 review response) Limitations para added, possible additions from appendices to main text?**
>
> Thank you for your thoughtful feedback. We address your concerns below.
>
> * “Beyond the method’s explanation, I found the experiment section to be poorly structured. The figures are small and difficult to follow – looking at all the figures it felt that “more is actually less”. Many of the evidence required to understand the method are only included in the appendices. However, having spent the time to go back and forth, I believe the experiments to be scientifically sound and convincing.”
>
> Thank you for taking the time to go through the appendices to understand our submission better, and apologies for having made this necessary. While we tried to keep the main manuscript concise, this was intended to aid readability and comprehension rather than hinder it. Is there any particular information you would suggest as particularly important to move up from appendices to the main text? We could also use some part of the remaining space to Figs 5, 6, 7, 8, 9 (showing prediction results from various methods) larger. Accounting for changes after other reviewer feedback, we now have about 1.2 pages left to reach the 10-page maximum.
>
> ---------------------------------------------
> * “I would have liked a discussion in the conclusion on the method’s limitation. This reviewer believes that the approach will struggle to deal with cyclic motions. In this case the discovered bottlenecks might not be the most useful to predict, as these will correspond to future frames (not nearby though) that are visually similar to the start (in forward) or to the start/end (in bidirectional) frames. ”
>
> Thank you for this suggestion. We have added a discussion of limitations to Sec 5, including this point. Yes, TAP in cyclic cases may not work in its current form. In particular, we expect that TAP will converge to the easy solution of repeating the input frame, since it will recur in the course of a cyclic motion. One solution is to use $\mathcal{E}’$ in the generalized minimum formulation of Sec 3.2 to express a preference for predicting frames that are different from the input frames. Another possibility is to preprocess states in some way to get TAP to work meaningfully. For instance, if the state is modified by appending the state visitation count, then the “cyclicity” of the trajectory would be destroyed so that bottlenecks are once again meaningful.
>
> ---------------------------------------------
> * “An additional loss to reward difficult-to-predict frames (though certain compared to other times) might be an interesting additional to conquer more realistic (rather than synthetic) video sequences.”
>
> Our generalized minimum formulation (Sec 3.2) already permits expressing additional objectives apart from ease of prediction. For instance, when we set $w(t)$ in future frame prediction to increase linearly with time as in Sec 4, we are indicating that while farther-away frames may be harder to predict, we would still prefer to predict those, so long as the prediction error is not too high. The idea of preferring predictions that are different from the input frames may also help address this point (cf. response to question about cyclicity above). Please also note that we already show results on BAIR pushing videos (non-synthetic video sequences), but we agree that results on videos from a broader domain would be interesting.

---

### Official Review · AnonReviewer2 · 2018-11-02
**simple yet effective**

**Rating:** 7
**Confidence:** 3

**Review:**

The authors present a method on prediction of frames in a video, with the key contribution being that the target prediction is floating, resolved by a minimum on the error of prediction. The authors show the merits of the approach on a synthetic benchmark of object manipulation with a robotic arm.

Quality: this paper appears to contain a lot of work, and in general is of high quality.

Clarity: some sections of the paper were harder to digest, but overall the quality of the writing is good and the authors have made efforts to present examples and diagrams where appropriate. Fig 1, especially helps one get a quick understanding of the concept of a `bottleneck` state.

Originality: To the extent of my knowledge, this work is novel. It proposes a new loss function, which is an interesting direction to explore.

Significance: I would say this work is significant. There appears to be a significant improvement in the visual quality of predictions. In most cases, the L1 error metric does not show such a huge improvement, but the visual difference is remarkable, so this goes to show that the L1 metric is perhaps not good enough at this point.

Overall, I think this work is significant and I would recommend its acceptance for publication at ICLR. There are some drawbacks, but I don’t think they are major or would justify rejection (see comments below).


I’m curious as to why you called the method in section 3.2 the “Generalized minimum”? It feels more like a weighted (or preference weighted) minimum to me and confused me a few times as I was reading the paper (GENerative? GENeralized? what’s general about it?). Just a comment.

What results does figure 4 present? Are they only for the grasping sequence? Please specify.

In connection with the previous comment, I think the results would be more readable if the match-steps were normalized for each sequence (at least for Figure 4). There would be a clearer mapping between fixT methods and the normalized matching step (e.g., we would expect fix0.75 to achieve a matching step of 0.75 instead of 6 / ? ).

Section 4, Intermediate prediction. The statement “the genmin w(t) preference is bell-shaped” is vague. Do you mean a Gaussian? If so, you should say “a Gaussian centered at T/2 and tuned so that …”

Section 4, Bottleneck discovery frequency. I am not entirely convinced by the measuring of bottleneck states. You say that a distance is computed between the predicted object position and the ground-truth object position. If a model were to output exactly the same frame as given in context, would the distance be zero? If so, doesn’t that mean that a model who predicts a non-bottleneck state before or after the robotic arm moves the pieces is estimated to have a very good bottleneck prediction frequency? I found this part of the paper the hardest to follow and the least convincing. Perhaps some intermediate results could help prospective readers understand better and be convinced of the protocol’s merits.



Typos:

Appendix E, 2nd paragraph, first sentence: “... generate an bidirectional state” --> “generate A bidirectional state”

---

> ### Author Response · Authors · 2018-11-14
> **(R2 review response) Clarifications and pdf updates**
>
> Thank you for your detailed questions and suggestions. We address your concerns below.
> -------------------------------------------------------------------------
> * "What results does figure 4 present? Are they only for the grasping sequence? Please specify. "
>
> Yes, Fig 4 is only for the grasping sequence. As stated in the para under “Forward prediction” on Pg 6, this is a scatter plot of minimum l1 error versus the closest-matching step for various models. We have changed the caption to present this information near the figure.
>
> -------------------------------------------------------------------------
> * "In connection with the previous comment, I think the results would be more readable if the match-steps were normalized for each sequence (at least for Figure 4). There would be a clearer mapping between fixT methods and the normalized matching step (e.g., we would expect fix0.75 to achieve a matching step of 0.75 instead of 6 / ? )."
>
> Thank you for pointing this out. We agree this does aid readability. We now present normalized match-step in both the figure as well as the table, for uniformity.
>
> -------------------------------------------------------------------------
> * “The statement “the genmin w(t) preference is bell-shaped and varies from 2/3 at the ends to 1 at the middle frame” is vague.”
>
> Agreed that this information should be more clearly presented. We omitted this in the submission to save space, but have included this in Appendix E now. In our experiments, w(t) was constructed as follows: the weight would rise linearly from baseval=0.66 at the first frame to 1.0 at the fifth frame, then stay at 1.0 for T-10 frames. From the (T-5)-th frame, it would drop linearly to baseval once more at the last frame. The only hyperparameter we tuned was baseval (search over 2/3 and 1/3).
>
> -------------------------------------------------------------------------
> * “Section 4, Bottleneck discovery frequency. I am not entirely convinced by the measuring of bottleneck states. You say that a distance is computed between the predicted object position and the ground-truth object position. If a model were to output exactly the same frame as given in context, would the distance be zero? If so, doesn’t that mean that a model who predicts a non-bottleneck state before or after the robotic arm moves the pieces is estimated to have a very good bottleneck prediction frequency? I found this part of the paper the hardest to follow and the least convincing.”
>
> A model that output the same frame as given in context would actually incur a heavy error. The distance is computed between the predicted object positions and the ground truth object positions *when exactly one of the two objects has been moved*, which may be reasonably assumed  to be the “bottleneck state” in this task. This means that outputting the starting context frame or the ending context frame would both produce heavy distance errors: the entire displacement of the first object, or the entire displacement of the second object, respectively.
>
> Unfortunately, the details of this measurement are quite involved, so we were forced to relegate this to Appendix E (now Appendix F after revisions). We have now added the example suggested by your comment to the paragraph in Section 4, to serve as an intuitive representative of the behavior of our method. Please let us know if this helps make things clearer.
>
> ----------------------------------------------------------------------------
> * "I’m curious as to why you called the method in section 3.2 the “Generalized minimum”? It feels more like a weighted (or preference weighted) minimum to me and confused me a few times as I was reading the paper (GENerative? GENeralized? what’s general about it?). Just a comment."
>
> The generalization here has to do with the relationship between Eq 4 and 5 in the submission. To call it “generalized minimum” is indeed not the most precise since there may be many other ways to generalize the minimum operator, but we choose to call it this for want of a better, concise term.
>
> Here is the case for considering it a generalization of the minimum. A standard minimum over i of a function f(i) can be written as: min_i f(i) = f({argmin_i f(i)}), as in Eq 4 in the submission.
>
> Now note that there are two occurrences of f(i) in the RHS expression above. The “generalized minimum” of Eq 5 generalizes this by allowing those two functions to be different as long as they are defined over the same domain:  genmin_i f(i) = f({argmin_i g(i)}).
>
> Restating in words, the standard minimum value of a function may be defined as the value of the function at *its own argmin*. Instead, the *generalized* minimum of a function f(.) with respect to a function g(.), is the value of f(.) evaluated at *the argmin of g(.)*. This is how “genmin” generalizes “min.”
>
> Please also take a look at the updated pdf portions and let us know if you have any further comments. Thank you.

---

### Public Comment · (anonymous) · 2018-11-11
**Two suggestions: (1) Add an experiment on real videos (e.g. human action) and (2) upload sample videos for the BAIR pushing dataset on the supplementary website.**

This approach seems simple as well as effective for me. However, I have two suggestions to improve the message of this paper.

First of all, this paper uses few robot simulations and the BAIR pushing (robot arm) dataset, which are relatively easy to memorize/predict compared to the real-world videos. Thus, to reassure this concern, I recommend the author to add results on human action dataset, like Human 3.6M or KTH (if this model takes a long time to train)  for example. I believe such an experiment would strengthen this work.

Also, on the website ( https://sites.google.com/view/ta-pred ) shared in this paper, it only includes videos for the simulation task. So, I believe it would be helpful to understand the effectiveness of this work if the author shares the videos from the BAIR pushing (and the human action) dataset on the supplementary website also.

---

> ### Author Response · Authors · 2018-11-15
> **BAIR  pushing videos added**
>
> Thank you for these suggestions. We have now added BAIR pushing videos to the website.
>
> KTH has some of the cyclic structure that R1 references in their review, so it may not be a good fit for TAP out of the box (we are adding a note on this limitation in our conclusions section). We will nevertheless attempt this soon, after prioritizing official review responses.

---

### Public Comment · ~Alexander_Neitz1 · 2018-11-15
**Related work**

Very exciting work! It would be great if you could include a brief comparison to the method proposed in the paper "Adaptive Skip Intervals: Temporal Abstraction for Recurrent Dynamical Models" (Neitz et. al, NIPS 2018; https://arxiv.org/abs/1808.04768 ).

---

> ### Author Response · Authors · 2018-11-16
> **Adding a para pointing out as concurrent work**
>
> Thanks for bringing this to our notice, and congrats on the great paper! We have added a brief paragraph in related work pointing this out as concurrent work with similar ideas.
> > "Concurrently with us, Neitz et al 2018 also propose a similar idea that allows a predictor to select when to predict, and their experiments demonstrate its advantages in specially constructed tasks with clear bottlenecks. In comparison, we propose not just the basic time-agnostic loss (Sec 3.1), but also improvements in Sec 3.2 to 3.4 that allow time-agnostic prediction to work in more general tasks such as synthetic and real videos of robotic object manipulation. Our experiments also test the quality of discovered bottlenecks in these scenarios and their usefulness as subgoals for hierarchical planning."

---

### Meta-Review · Area_Chair1 · 2018-12-12
**solid work, would merit from more experimentation**

**Confidence:** 5
**Recommendation:** Accept (Poster)

**Metareview:**

The paper introduces a new and convincing method for video frame prediction, by adding prediction uncertainty through VAEs.  The results are convincing, and the reviewers are convinced.

It's unfortunate however that the method is only evaluated on simulated data.  Letting it loose on real data would cement the results and merit oral representation; in the current form, poster presentation is recommended.

---

> ### Author Response · Authors · 2018-12-21
> **Clarifying contribution and real-data experiments**
>
> Thank you, we are glad to have the opportunity to present our work at ICLR 2019!
>
> A couple of clarifications for interested readers:
> (i) This paper's contribution is not about maintaining prediction uncertainty through VAEs. Instead, the idea is to allow predictors to select which timesteps to make predictions about. We show that this not only improves prediction quality but also consistently predicts semantically coherent changepoints that can be used, for instance, as subgoals for planning.
> (ii) We do in fact have results for real videos (the BAIR pushing dataset) both in our paper and on the website.
>
> Thank you,
> (On behalf of the authors)